# *Model X-ray* : Detecting Backdoored Models via Decision Boundary

## ABSTRACT

Backdoor attacks pose a significant security vulnerability for deep neural networks (DNNs), enabling them to operate normally on clean inputs but manipulate predictions when specific trigger patterns occur. In this paper, we consider a practical post-training scenario backdoor defense, where the defender aims to evaluate whether a trained model has been compromised by backdoor attacks. Currently, post-training backdoor detection approaches often operate under the assumption that the defender has knowledge of the attack information, logit output from the model, and knowledge of the model parameters, limiting their implementation in practical scenarios. In contrast, our approach functions as a lightweight diagnostic scanning tool that operates in conjunction with other defense methods, assisting in defense pipelines.

We begin by presenting an intriguing observation: the decision boundary of the backdoored model exhibits a greater degree of closeness than that of the clean model. Simultaneously, if only one single label is infected, a larger portion of the regions will be dominated by the attacked label. Leveraging this observation, drawing an analogy to X-rays in disease diagnosis, we propose *Model X-ray* . This novel backdoor detection approach is based on the analysis of illustrated two-dimensional (2D) decision boundaries, offering interpretability and visualization. *Model X-ray* can not only identify whether the target model is infected but also determine the target attacked label under the all-to-one attack strategy. Importantly, it accomplishes this solely by the predicted hard labels of clean inputs, regardless of any assumptions about attacks and prior knowledge of the training details of the model. Extensive experiments demonstrated that *Model X-ray* can be effective and efficient across diverse backdoor attacks, datasets, and architectures.

## CCS CONCEPTS

• **Security and privacy**; • **Computing methodologies** → **Machine learning**;

## KEYWORDS

Deep Learning, Backdoor Detection, Decision Boundary

## 1 INTRODUCTION

Despite the remarkable success of DNNs, recent studies [6, 13, 19, 29, 33, 45, 50] have unveiled a significant security vulnerability

*ACM MM, 2024, Melbourne, Australia*

© 2024 Copyright held by the owner/author(s). Publication rights licensed to ACM.
ACM ISBN 978-x-xxxx-xxxx-x/YY/MM
https://doi.org/10.1145/nnnnnnn.nnnnnnn

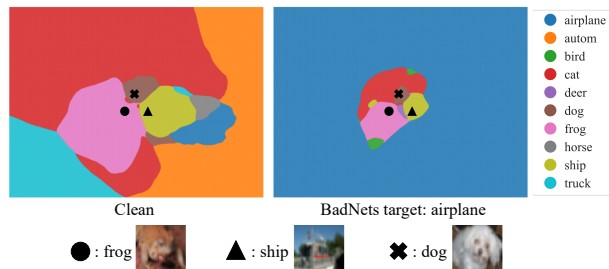

**Figure 1: Comparison of the decision boundaries between the clean model and the backdoored model (taking BadNets [19] as an example, and the target label is "airplane") on the CIFAR-10 dataset.**

for DNNs against backdoor attacks, which can contaminate DNNs, enabling them to operate normally on clean inputs but manipulate predictions when specific patterns (*i.e.*, "trigger") occur. Backdoor attacks primarily fall into two categories: data-poisoning attacks (such as BadNets [19], SSBA [29], Low Frequency [50], and BPP [45]) and model-modification attacks (such as TrojanNN [33], LIRA [13], and Blind [6]). These attacks pose a substantial threat to safety-critical and security-sensitive applications of DNNs, including but not limited to face recognition [35], biomedical diagnosis [16], and autonomous driving [37]. To mitigate the threat of backdoor attacks, numerous defense methods are emerging to establish a comprehensive pipeline for backdoor defense. This pipeline can be applied at various stages, including the training, post-training, and deployment stages (refer to Fig. 2).

Backdoor defense during both the training and deployment stages [8, 18, 31, 41, 50] typically necessitates access to training data or inference data. In this paper, we consider the more practical post-training scenario, where the defender aims to evaluate whether a trained model (*e.g.*, Model Zoo that provides pre-trained models [1]) has been compromised by backdoor attacks, when and many post-training defenses assume the defender independently possesses a small set of clean, legitimate samples. However, current post-training detection methods hold too strong assumptions that the defender has knowledge of the attack information, the logit output from the model [9, 49], and knowledge of the model parameters [17, 32, 42, 43], limiting their implement in practical scenarios.

Fortunately, recent work by [39] has demonstrated that we can visualize the model's decision boundary solely using prediction labels. Leveraging this technique, we have identified a discernible distinction between the decision boundaries of the clean model and the backdoored model. As illustrated in Fig. 1, we use BadNets [19] as an example of backdoor attacks. We observe that the decision boundaries of backdoor models exhibit a noticeable reduction in the regions dominated by three clean samples, and significant surrounding area are dominated by the attack target label. Importantly, this phenomenon is applicable across various backdoor attacks on

 

different datasets (see Fig. 4). That is to say, we can leverage the phenomenon of anomalous decision boundaries to distinguish backdoored models. As claimed in previous work [42], backdoor attacks build a shortcut leading to the target label, which we explain cause the above encircling phonemena. Besdies, trigger samples are more robust against distortions [36], causing the large regions than that of clean samples. In a nutshell, the visualized 2D decision boundary can be served as an illustration for these conjectures.

Based on the intriguing phenomenon, drawing an analogy to X-rays in disease diagnosis, we propose *Model X-ray* as a novel backdoor detection approach through the analysis of illustrated 2D decision boundaries. Specifically, we designate two metrics to evaluate the degree of the closeness of the decision boundary: 1) **Rényi Entropy (RE)** [38] calculated on the probability distribution of each prediction area and 2) **Areas Dominated by triple samples (ATS)**, *e.g.*, the total areas of "frog", "ship", and "dog" in the Fig. 1. Furthermore, if only one label is infected, we can determine the target label by the prediction of the largest area of the decision boundary, *e.g.*, the target label is "airplane" in the right of Fig. 1. In other words, *Model X-ray* can not only identify backdoored models but also determine the target attacked label under all-to-one attacks. Importantly, *Model X-ray* accomplishes this only by the predicted hard labels of clean inputs from the model, regardless of any assumptions about attacks such as the trigger patterns and training details. The visualized 2D decision boundary offers a novel perspective to understand the behavior of the model, providing both visualization and interpretability. Through analysis of the decision boundary, *Model X-ray* can function as a lightweight diagnostic scanning tool, complementing other defense methods and aiding in defense pipelines. Extensive experiments demonstrate that *Model X-ray* performs better than current methods across various backdoor attacks, datasets, and model architectures. In addition, some ablation studies and discussions are also provided.

Our contributions can be summarized as follows:

- We present a noteworthy observation: there exists a distinction between clean models and backdoored models by visualized 2D decision boundaries [39].
- We propose *Model X-ray* which detects the backdoored model solely by predicted hard labels of clean inputs from the model, regardless of any assumptions about backdoor attacks. Besides, *Model X-ray* can determine the target attacked label if the attack is all-to-one attack.
- Extensive experiments demonstrate the effectiveness and efficiency of *Model X-ray* across different backdoor attacks, datasets, and model architectures.

## 2 RELATED WORK

### 2.1 Backdoor Attacks

The target of backdoor attacks is training an infected model $\hat{M}$ with parameters $\theta$ by:

$$\theta = \arg\min_{\theta} \mathbb{E}_{(x,y)\sim\mathcal{D}} \mathcal{L}(\hat{M}(x;\theta), y) \\ + \mathbb{E}_{(\hat{x},y_t)\sim\hat{\mathcal{D}}} \mathcal{L}(\hat{M}(\hat{x};\theta), y_t), \quad (1)$$

where $\mathcal{D}$ and $\hat{\mathcal{D}}$ denote the benign samples and trigger samples, respectively. $\mathcal{L}$ denotes the loss function, *e.g.*, cross-entropy loss. The infected model functions normally on benign samples but yields a specific target prediction $y_t$ when presented with trigger samples $\hat{x}$. Backdoor attacks can be achieved by data poisoning and model modification, and we briefly introduce some related methods below.

Data poisoning-based backdoor attacks primarily revolve around crafting trigger samples. Notably, BadNets [19] was a pioneering work that highlighted vulnerabilities of DNNs by employing visible squares as triggers. Afterward, various other visible trigger techniques have been explored: Blended [10] employs image blending to create trigger patterns, SIG [7] utilizes sinusoidal strips as triggers, and Low Frequency (LF) [50] explores triggers in the frequency domain. Simultaneously, other research endeavors focus on achieving imperceptibility of the trigger patterns, including BPP [45] based on image quantization and dithering, WaNet [34] founded on image warping, and SSBA [29] achieved by image steganography. During the training stage, the attacker can leverage different poisoning ratios to balance the attack ability and performance degradation.

Apart from data poisoning-based attacks, there are some backdoor attacks that employ model modification techniques. TrojanNN [33] first proposes to optimize the trigger to ensure that the crucial neurons can attain their maximum values, LIRA [13] formulates malicious function as a non-convex, constrained optimization problem to learn invisible triggers through a two-stage stochastic optimization procedure, and Blind [6] modifies the training loss function to enable the model to learn the malicious function.

### 2.2 Backdoor Defenses

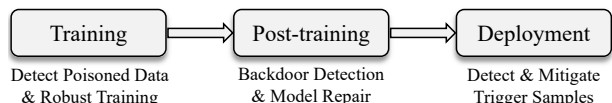

**Figure 2: The pipeline of the backdoor defense.**

As Fig. 2 illustrates, pipelines for backdoor defense mechanisms can be categorized into three phases: during training, post-training, and after deployment. Each phase implies distinct defender roles and capabilities.

Backdoor defenses during model training aim to detect and remove poisoned data from the training set [8, 40, 41] or to enhance training robustness against data poisoning [30]. Backdoor defenses after deployment aim to detect trigger inputs during inference and attempt to mitigate the malicious prediction. For example, STRIP[18] perturbs an input sample by overlapping with numerous benign samples and uses the ensemble predictions for detection. FreqDetector [50] leverages artifacts in the frequency domain to distinguish trigger samples from clean samples. Besides, some methods [21, 31, 36] conduct detection based on robustness against data transformations between benign and trigger samples.

Comparably, post-training backdoor detection is model-level detection. Neural Cleanse [42] is the first post-training detection through anomaly analysis on the reversed trigger patterns. However, it requires access to the model's inner information like parameters and gradients, which is also the limitation of other subsequent methods [17, 32, 42–44]. Differently, detection work in black-box scenarios is extremely challenging [9, 14, 20, 49], *e.g.* MNTD trains a meta-classifier based on features extracted from a large set of shadow models. However, its success heavily relies on the generalization capability of the attack settings from the shadow models

to the actual backdoored models. Besides, it requires the soft label generated by the target model. MM-BD [43] leverags maximum margin statistics of each class and unsupervised anomaly detection on classifier output landscapes.

## 2.3 Decision Boundary of Deep Neural Networks

Most previous works depict decision boundaries by adversarial samples [23, 27] or sensitive samples [24]. These methods are pivotal in identifying and understanding the contours of decision boundaries, as adversarial and sensitive samples are typically positioned along these critical junctures in the model's decision-making process. However, obtaining these special samples requires access to the target model. Fortunately, Zhang *et al.* [51] find that decision boundaries not only manifest near the data manifold but also within the convex hull created by pairs of data points.

Leveraging this understanding, Somepalli *et al.* [39] introduce an innovative approach that utilizes only clean samples to map out the decision boundary to investigate reproducibility and double descent. Their method, which results in a 2D map, offers an intuitive and accessible means of visualizing decision boundaries. In this paper, we utilize this technique to detect backdoored models.

## 3 PRELIMINARIES

### 3.1 Recap of the Decision Boundary in [39]

Here, we recap the methods for visualizing decision boundaries discussed in [39]. As shown in Fig. 3 (left), we randomly choose three clean samples (also called **triple samples**) from the dataset $\mathcal{D}$. For example, we select three images $(x_1, x_2, x_3)$ of "frog", "ship", and "dog" from the CIFAR-10 dataset. Then, we can calculate two vectors $\vec{v_1} = x_2 - x_1$ and $\vec{v_2} = x_3 - x_1$, based on which we obtain the spanned space $\mathcal{V}$, *i.e.*, $\mathcal{V} = span\{\vec{v_1}, \vec{v_2}\}$, whose orthogonal basis and orthonormal basis are denoted as $\{\vec{\beta_1}, \vec{\beta_2}\}$ and $\{\vec{e_1}, \vec{e_2}\}$, respectively, where $\vec{\beta_1} = \vec{v_1}$ and $\vec{e_1} = \frac{\vec{\beta_1}}{\|\vec{\beta_1}\|}$ . Next, we can obtain the projection of vector $\vec{v_2}$ in the direction of vector $\vec{e_1}$, *i.e.*, $\mathrm{proj}_{\vec{e_1}}\vec{v_2} = \langle\vec{v_2}, \vec{e_1}\rangle \cdot \vec{e_1}$ and get $\vec{\beta_2}$ by orthogonalizing $\vec{v_2}$ via Schmidt orthogonalization, *i.e.*, $\vec{\beta_2} = \vec{v_2} - \mathrm{proj}_{\vec{e_1}}\vec{v_2}$. Similarly, we can acquire the projection of vector $\vec{v_2}$ in the direction of vector $\vec{e_2}$, *i.e.*, $\mathrm{proj}_{\vec{e_2}}\vec{v_2} = \langle\vec{v_2}, \vec{e_2}\rangle \cdot \vec{e_2}$. Finally, we obtain an orthonormal basis for the space, denoted as $\vec{e_1}$ and $\vec{e_2}$, along with the coordinates of points $x_1$, $x_2$, and $x_3$ within the plane. Namely, we acquire coordinates corresponding to the origin $(0, 0)$ and the points specified by vectors $\vec{v_1}$ and $\vec{v_2}$, originating from the origin, i.e., $(0, 0)$, $(\|\vec{v_1}\|, 0)$, $(\mathrm{proj}_{\vec{e_1}}\vec{v_2}, \mathrm{proj}_{\vec{e_2}}\vec{v_2})$.

After representing the space, we can calculate the bounds on the X-axis and the Y-axis, extended by a factor of $\eta$ in both the positive and negative directions along the corresponding axes, serving as a means to control the expansion range of the coordinate system. In the previous work [39], $\eta$ is set as 1 to investigate reproducibility, while we set $\eta$ as 5 to obtain a wider range of the decision boundary (see Fig. 3). Moreover, we can also determine density $S$ by constructing the set of points with a quantity of $S^2$ within the bounded range of the coordinate system using a grid generation

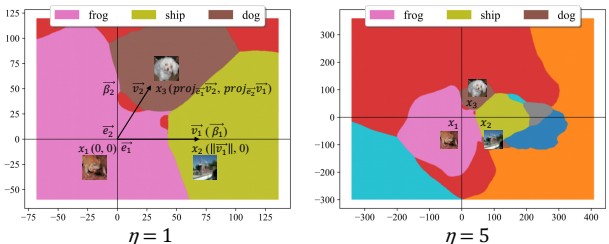

**Figure 3: Visual examples of the decision boundary used in [39] (left) and in this paper (right).**

method. Larger $S$ means higher resolution. With $S^2$ points, we can conduct the reverse process to get their tensor presentation, which can be fed to the model to fetch the corresponding prediction. We adopt different colors for different predictions to get the final 2D decision boundary.

In the subsequent parts, all decision boundaries are visualized by the modified version (*i.e.*, $\eta = 5$ in the right of Fig. 3).

### 3.2 Threat Model

In practice, acccess to training or inference sets is unavailable due to data privacy, ownership, and availability constraints. Therefore, in this paper, we only consider the post-training scenario detection.

While many post-training detection defenses typically have access to either the model's weights[17, 32, 42–44] or the model's logit output[49] for evaluation, our approach goes a step further by restricting access to the model. We only assume that the defender has the capability to independently gather a small set of clean data samples that cover all classes within the domain, a prerequisite upon which most post-training detectors depend. Moreover, we only need the hard label predictions of the target model.

## 4 METHOD

In this section, we first provide an intriguing observation on the decision boundary of clean models and backdoor models. Based on this, we designate two strategies for backdoor detection via the decision boundary. Finally, we showcase that we can determine the target attacked label, if only one single label is infected.

### 4.1 An Intriguing Observation

As shown in Fig. 4, we provide the decision boundary of the clean model and different backdoored models (infected by BadNets [19], SSBA [29], LF [50], BPP [45], TrojanNN [33], LIRA [13], and Blind [6]) on CIFAR-10 and ImageNet-10 dataset. We observe that the decision boundaries of backdoor models exhibit a noticeable reduction in the area of decision regions dominated by three clean samples, and significant surrounding area are dominated by the attack target label, *i.e.*, the phenomenon of anomalous decision boundaries. Therefore, the label distribution within the decision boundaries of the backdoor model becomes highly concentrated, exhibiting the attack target label with an abnormally high probability. More visualized decision boundaries can be found in the supplementary material.

We explain this phenomenon may be the shortcut effect caused by backdoor attacks. In essence, clean models can still preserve the

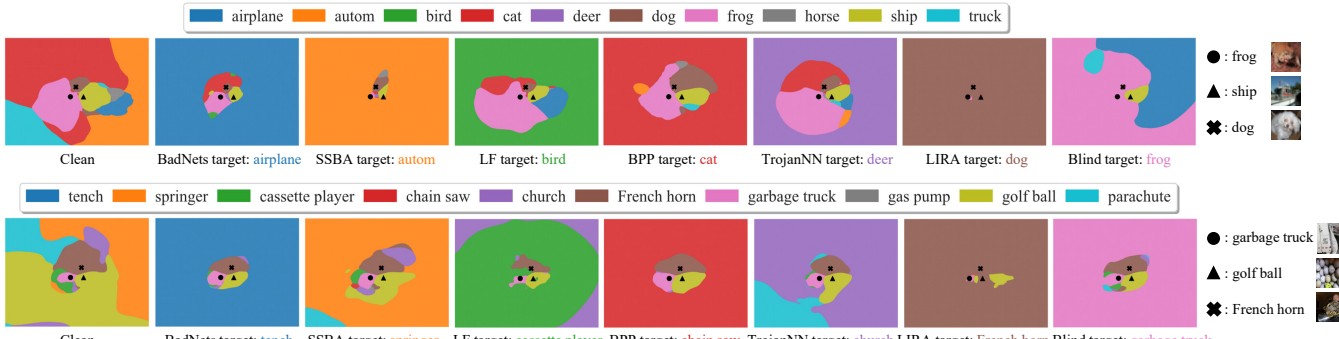

Figure 4: Visual examples of decision boundaries of the clean model and different backdoored models on CIFAR-10 and ImageNet-10.

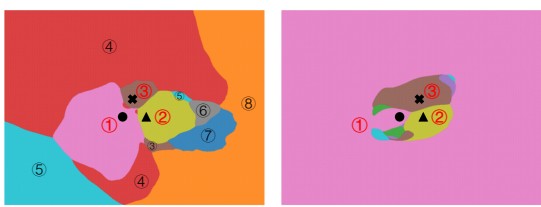

Figure 5: Illustration on calculation of RE and ATS.

robustness of predicted labels when applying a linear transformation to samples in a considerably large magnitude. On the contrary, the presence of shortcuts to the target label in backdoor models results in changes in the predicted label when applying a minor linear transformation to samples, typically leading to the target attacked label. The shortcuts leading to the target attacked label in the backdoor model has been confirmed in previous research, that is, through optimization methods, smaller perturbations can be found to cause other labels to be misclassified as target labels [42]. Afterward, Rajabi *et al.* [36] quantifies this effect by introducing the concept of a certified radius [11], which estimates the distance to a decision boundary by perturbing samples with Gaussian noise with a predetermined mean and variance. Notably, trigger samples are observed to be relatively farther from a decision boundary compared to clean samples, which can support why the large region is dominated by injected prediction.

## 4.2 Two Strategies for Backdoor Detection via the Decision Boundary

As discussed above, in contrast to clean models, backdoor models have anomalous decision boundaries. Therefore, backdoor detection can be transformed into anomaly detection on the decision boundary. To achieve this, we propose two strategies for backdoor detection via the decision boundary, namely, based on **Rényi Entropy (RE)** and **Areas dominated by Triple Samples (ATS)**, respectively. In the following part, we will introduce the two strategies in detail, which we hope sheds some light on anomaly detection. Notably, other strategies are also applicable.

*4.2.1 Backdoor Detection based on Rényi Entropy.* With the technique mentioned above, we can plot $N$ decision boundaries $\mathbf{B} = \{\mathcal{B}_1, ..., \mathcal{B}_k, ..., \mathcal{B}_N\}$, where $\mathcal{B}_k$ is plotted along the plane spanned by triple samples $T_k = (x_1, x_2, x_3)_k$. Specifically, let $S_k = \{x_{ij}|(i, j) \in$

$\mathcal{B}_k\}$ be the set of points in the $\mathcal{B}_i$, where $(i, j)$ is the coordinations of $x$ in $\mathcal{B}_k$. Then, we feed $S_k = \{x_{ij}|(i, j) \in \mathcal{B}_k\}$ to the target model $M$ to obtain the corresponding hard labels $L_k = \{l_{ij}|(i, j) \in \mathcal{B}_k\}$, which are further used to obtain the final colorful decision boundary $\mathcal{B}_k$ for evaluation.

Within a specific decision boundary $\mathcal{B}_k$, we calculate *label probability distribution* $\mathcal{P}_k = \{p_1, ..., p_m, ..., p_n\}$ for n-category classification:

$$p_m = \frac{A(l_m)}{A(\mathcal{B}_k)}, \tag{2}$$

where $l_m$ denotes the $m$-th class label in the dataset. $A(l_m)$ and $A(\mathcal{B}_k)$ denote the areas of $m$-th class and the areas of entire decision regions, respectively. In Fig. 5 (left), $p_3 = (A(③) + A(③))/A(\mathcal{B}_k)$. To indirectly evaluate the gathering degree of the decision boundary, we calculate **Rényi Entropy (RE)** of label probability distribution $\mathcal{P}_k$:

$$RE(\mathcal{P}_k) = H_\alpha(\mathcal{P}_k) = \frac{1}{1-\alpha} \log\left(\sum_{m=1}^n p_m^\alpha\right), \tag{3}$$

where $\alpha \geq 1$, and we set it as 10 by default. Based on **RE**, we propose a detection strategy called **Ours-RE**. Briefly, a large variance of $\{p_1, ..., p_m, ..., p_n\}$ will lead a low **RE**, meaning more gathered. As shown in Fig. 4, we find backdoored models hold much lower **RE**, which can be distinguished from the clean model in most cases.

*4.2.2 Backdoor Detection based on Areas dominated by Triple Samples.* In addition to **RE**, we define **Areas dominated by Triple Samples (ATS)** as the ratio of decision regions controlled by benign triple samples $T_k$ to entire decision regions:

$$ATS(\mathcal{B}_k) = \frac{A(T_k)}{A(\mathcal{B}_k)} = \frac{\sum_{x \in (x_1, x_2, x_3)} A(x)}{A(\mathcal{B}_k)}, \tag{4}$$

where $A(T_k)$ denotes the total areas dominated by triple samples. As shown in the left of Fig. 5, $ATS(\mathcal{B}_k) = (A(①) + A(②) + A(③))/A(\mathcal{B}_k)$. However, we find there are some special cases. As shown in Fig. 5 (right), one of the triple samples belongs to the target attacked label, causing an abnormally large $A(①)$. In practice, we cannot determine whether the labels of triple samples are injected. For this, we append an additional constraint for **ATS**, namely, $A(x) < A(\mathcal{B}_k) \cdot t$, where $t = 0.5$ by default. Based on **ATS**, we propose a detection strategy called **Ours-ATS**. Intuitively, the large **ATS** means robust classification on the clean images, and vice versa.

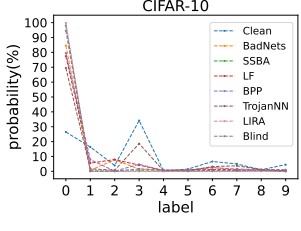
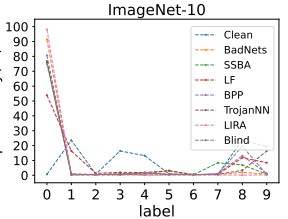

**Figure 6: The label probability distribution within decision boundaries of clean and backdoor models on CIFAR-10 and ImageNet-10, both of whose infected labels are 0.**

## 4.3 Determine the Target Label

After detecting, if the attack is conducted by all-to-one strategy, defenders can further determine the target attacked label by identifying the label with an abnormally high probability in label probability distribution $\mathcal{P}_k = \{p_1, ..., p_m, ..., p_n\}$. For example, we plot decision boundaries of clean models and backdoor models infected by different backdoor attacks on CIFAR-10 and ImageNet-10 datasets. For each model, we plot 20 decision boundaries and calculate the average label probability. As shown in Fig. 6, the attacked target label (label "0" of both CIFAR-10 and ImageNet-10) exhibits an exceptionally high probability, even reaching 80% to 90% of the entire label probability distribution.

## 5 EXPERIMENT

### 5.1 Experimental Settings

**Datasets and Architectures.** The datasets include CIFAR-10 [28], CIFAR-100 [28], GTSRB [25], and ImageNet-10 [2], a subset of ten classes from ImageNet [12]. Besides, we employ four different architectures: PreActResNet-18 [22], MobileNet-V3-Large [26], PreActResNet-34 [22], and ViT-B-16 [15]. These architectures encompass both Convolutional Neural Networks (CNNs) and Vision Transformers (ViTs) and span across various network sizes, including small, medium, and large networks.

**Implementation Details.** For the model to be evaluated, we plot decision boundaries by random samples triplet with expansion factor $\eta = 5$ and density $S = 100$, number of plots $N = 20$. For the attack baselines, we evaluate our method against seven backdoor attacks, including BadNets [19], SSBA [29], LF [50], BPP [45], TrojanNN [33], LIRA [13], and Blind [6]. We follow an open-sourced backdoor benchmark BackdoorBench [46] for the training settings of these attacks and conduct all-to-one attacks by default. As shown in Table 1, the attacks in our experiments include both data poisoning-based attacks and model modification-based attacks, which contain diverse and complex trigger pattern types. In this paper, our focus is on post-training backdoor detection. We compare our approach with three post-training detection methods: Neural Cleanse [42], MNTD [49], and MM-BD [43]. We utilize their official implementations [3, 5] or implementations available in open-source benchmarks[4].

**Evaluation Metrics.** For clean models and models infected by 7 backdoor attacks, we trained 20 models using different initialization and random seeds. For the backdoored models, we select different attack target labels and conduct the single-label attack by default. Considering the computational cost, we adopted different data

**Table 1: The backdoor attacks involved in our evaluations have covered diverse trigger patterns.**

| Trigger | Data Poisoning | | | | Model Modification | | |
|---|---|---|---|---|---|---|---|
| | BadNets | SSBA | LF | BPP | TrojanNN | LIRA | Blind |
| Static | ● | ○ | ● | ○ | ● | ○ | ● |
| Invisible | ○ | ● | ● | ● | ○ | ● | ○ |
| Dynamic | ○ | ● | ○ | ● | ○ | ● | ○ |

sets and corresponding common model architectures. Thus, we have $20 + 20 \times 7 = 160$ models for each combination of dataset and architecture. In subsequent experiments, for each model to be evaluated, we calculate its average **RE** (see Eq. (3)) and **ATS** (see Eq. (4)) over $N = 20$ decision boundary plots as indicators. We assume that defense mechanisms return a positive label if they identify a model as a backdoored model and then compute the *Area Under Receiver Operating Curve (AUROC)* to measure the trade-off between the *false positive rate (FPR)* for clean models and *true positive rate (TPR)* for backdoor models for a detection method.

### 5.2 The Effectiveness of *Model X-ray*

As shown in Table 2, in most cases, *Model X-ray* outperforms the baseline methods across different backdoor attacks, datasets, and architectures. MNTD is difficult to generalize attack settings from the shadow models to the actual backdoored models. Neural Cleanse performs well in the majority of scenarios. However, occasional failures may arise when it incorrectly identifies a trigger for a clean model, leading to convergence in local optima. MM-BD demonstrates promising performance on small-scale architectures, but its performance drops significantly on larger architectures. In Fig. 7 and Fig. 8, we present visual illustrations of the average **RE** and **ATS** values for both clean and backdoored models. In most cases, a clear distinction is evident between clean and backdoored models. The ROC curves of **Ours-RE** and **Ours-ATS** can be found in the supplementary material.

Besides the default all-to-one attack strategy, we consider attack strategies [48] with arbitrary numbers of source classes each assigned with an arbitrary attack target class, including X-to-X attack, X-to-one attack, and one-to-one attack. We adopt different attack strategies to conduct BadNets on CIFAR-10. For each strategy, we train 10 models for evaluation. Table 3 shows that *Model X-ray* remains effective under different attack strategies, especially based on **ATS** (*i.e.*, **Ours-ATS**). Although multi-target attacks lower the performance of the proposed method, we outperform the baseline methods by a large margin in most cases. Furthermore, we provide some visual examples of the corresponding decision boundary in Fig. 11. In X-to-one and one-to-one attacks, where the attack target is a single class, both **Ours-RE** and **Ours-ATS** achieve precise detection and identification of the target class. In X-to-X attack, where there are multiple classes for both source and attack targets, the performance of **Ours-RE** declines with an increasing number of attack target classes, which is acceptable. The computation of **Ours-RE** relies on the entropy of class labels, where it can still detect the presence of multiple attack target classes in the decision boundary, despite the performance drop. Furthermore, areas dominated by

**Table 2: The performance of *Model X-ray* across different attacks, datasets, and architectures. The last two columns show the worst and the average performance among different attacks. The best results are in bold.**

| Dataset
Architecture | Attack→
Method↓ | BadNets | SSBA | LF | BPP | TrojanNN | LIRA | Blind | Worst | Average |
|---|---|---|---|---|---|---|---|---|---|---|
| CIFAR-10 | Neural Cleanse | 0.881 | 0.755 | 0.874 | **0.881** | 0.566 | 0.884 | 0.535 | 0.535 | 0.768 |
| | MNTD | 0.525 | 0.665 | 0.568 | 0.565 | 0.568 | 0.623 | 0.705 | 0.525 | 0.603 |
| | MM-BD | **1.000** | 0.847 | **0.882** | 0.805 | **0.860** | 0.953 | 0.697 | 0.697 | 0.863 |
| PreActResNet-18 | Ours-RE | 0.995 | **1.000** | 0.812 | 0.762 | 0.740 | **1.000** | **0.919** | 0.740 | 0.890 |
| | Ours-ATS | **1.000** | **1.000** | 0.763 | 0.747 | 0.848 | **1.000** | 0.885 | **0.747** | **0.892** |
| GTSRB | Neural Cleanse | 0.997 | 0.968 | 0.937 | 0.965 | 0.661 | 0.715 | 0.990 | 0.661 | 0.890 |
| | MNTD | 0.603 | 0.495 | 0.578 | 0.617 | 0.535 | 0.715 | 0.460 | 0.460 | 0.572 |
| | MM-BD | 1.000 | 0.477 | 0.494 | 0.445 | 0.792 | 0.994 | 0.997 | 0.445 | 0.743 |
| MobileNet-V3 | Ours-RE | 0.997 | 0.981 | 0.942 | **1.000** | **0.976** | **1.000** | **1.000** | **0.942** | **0.985** |
| -Large | Ours-ATS | **0.998** | **0.997** | **0.972** | 0.982 | 0.902 | 0.996 | **1.000** | 0.902 | 0.978 |
| CIFAR-100 | Neural Cleanse | 0.975 | 0.882 | 0.811 | 0.807 | **0.970** | 0.970 | 0.700 | 0.700 | 0.874 |
| | MNTD | 0.625 | 0.490 | 0.540 | 0.528 | 0.540 | 0.813 | 0.538 | 0.490 | 0.582 |
| | MM-BD | 0.626 | 0.552 | 0.977 | 0.557 | 0.618 | 0.957 | 0.633 | 0.552 | 0.703 |
| PreActResNet-34 | Ours-RE | **1.000** | **1.000** | **1.000** | 0.832 | 0.746 | 0.979 | 0.819 | 0.746 | 0.911 |
| | Ours-ATS | **1.000** | **1.000** | **1.000** | **0.900** | **0.997** | **0.988** | **0.977** | **0.900** | **0.980** |
| ImageNet-10 | Neural Cleanse | 0.955 | 0.808 | 0.683 | 0.927 | 0.847 | 0.969 | 0.913 | 0.683 | 0.872 |
| | MNTD | 0.588 | 0.428 | 0.620 | 0.323 | 0.620 | 0.632 | 0.478 | 0.323 | 0.527 |
| | MM-BD | 0.107 | 0.205 | 0.135 | 0.120 | 0.215 | 0.518 | 0.149 | 0.107 | 0.207 |
| ViT-B-16 | Ours-RE | 0.956 | 0.860 | 0.835 | 0.913 | 0.725 | **1.000** | 0.863 | 0.725 | 0.879 |
| | Ours-ATS | **1.000** | **0.861** | **0.956** | **0.976** | **0.878** | **1.000** | **0.935** | **0.878** | **0.944** |

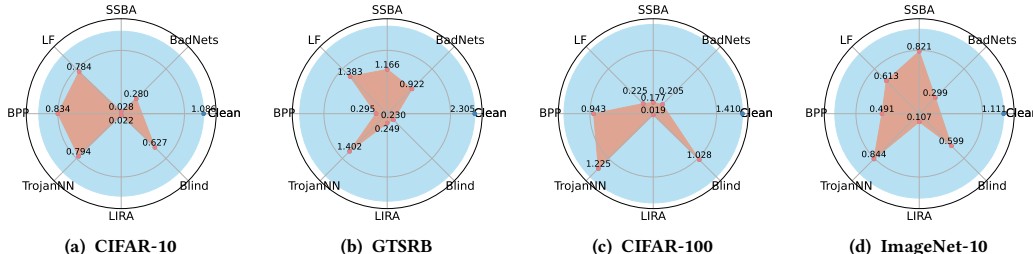

(a) CIFAR-10     (b) GTSRB     (c) CIFAR-100     (d) ImageNet-10

**Figure 7: The average RE ($\alpha = 10$) for clean and backdoor models injected by seven backdoor attacks in CIFAR-10, CIFAR-100, GTSRB, and ImageNet-10 datasets. We observe that backdoor models have significantly smaller RE than clean models.**

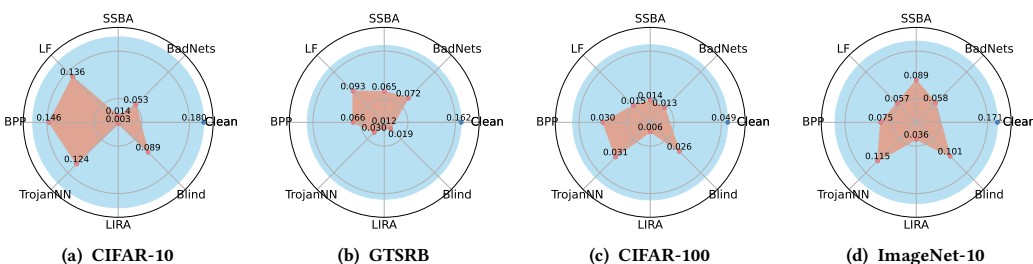

(a) CIFAR-10     (b) GTSRB     (c) CIFAR-100     (d) ImageNet-10

**Figure 8: The average ATS ($t = 0.5$) for clean and backdoor models injected by seven backdoor attacks in CIFAR-10, CIFAR-100, GTSRB, and ImageNet-10 datasets. We observe that backdoor models have significantly smaller ATS than clean models.**

**Table 3: The performance under different attack strategies.**

| Strategy | 10to1 | 5to1 | 2to1 | 1to1 | 3to3 | 5to5 | 10to10 |
|---|---|---|---|---|---|---|---|
| Neural Cleanse | 0.881 | 0.845 | 0.784 | 0.826 | 0.423 | 0.284 | 0.439 |
| MNTD | 0.525 | 0.419 | 0.503 | 0.487 | 0.535 | 0.518 | 0.466 |
| MM-BD | 1.000 | 0.571 | 0.006 | 0.081 | 0.007 | 0.448 | 0.671 |
| Ours-RE | **1.000** | **0.995** | 0.824 | 0.829 | **0.839** | 0.638 | 0.423 |
| Ours-ATS | **1.000** | **0.995** | **0.967** | **0.862** | 0.821 | **0.862** | **0.746** |

triple clean samples shrink, which explains why **Ours-ATS** achieves good performance in such scenarios.

## 5.3 Evaluations on Open-source Benchmarks

To mitigate the impact of incidental factors in our training, we also evaluated our method on the backdoored models pre-trained on

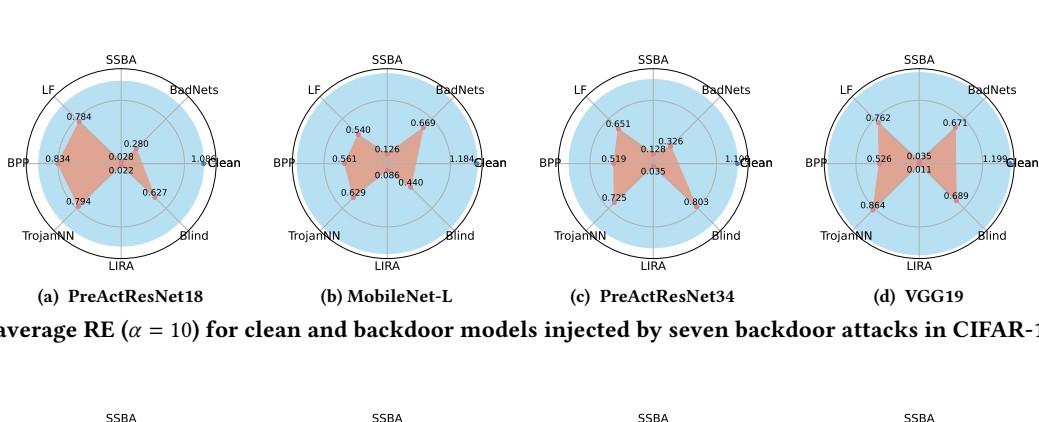

Figure 9: The average RE ($\alpha = 10$) for clean and backdoor models injected by seven backdoor attacks in CIFAR-10 on different architectures.

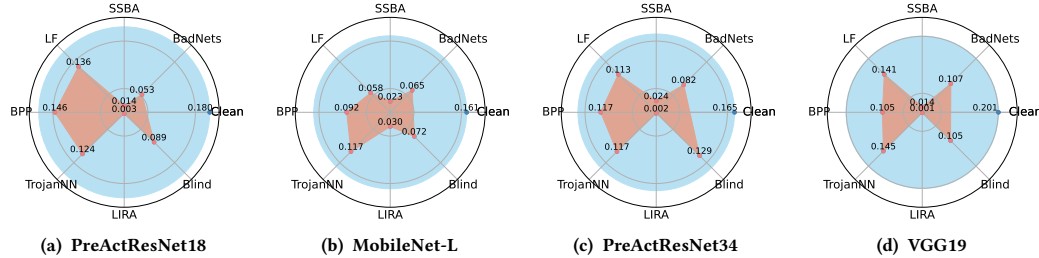

Figure 10: The average ATS ($t = 0.5$) for clean and backdoor models injected by seven backdoor attacks in CIFAR-10 on different architectures.

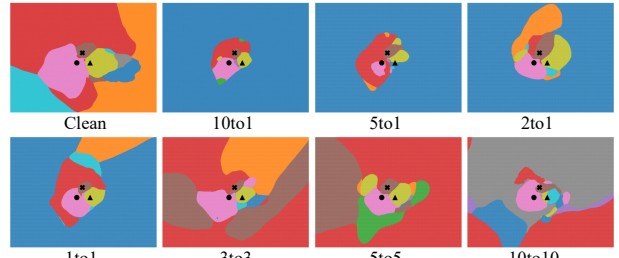

Figure 11: Decision boundaries under different attack strategies.

an open-source benchmark [46]. Speifically, we perform detection on pre-trained backdoored models injected with seven backdoor attacks across CIFAR-10, GTSRB, and CIFAR-100 datasets using the PreActResNet-18 architecture, which can be downloaded from Open-source benchmarks [4].

Given a target model $C_\theta$, *Model X-ray* map the model $C_\theta$ to a linearly separable space, defenders can make judgments through average **RE** and **ATS** based on a threshold $\gamma$ :

$$\Gamma(\text{Model X-ray}(C_\theta)) = \begin{cases} 1, \text{Model X-ray}(C_\theta) \leq \gamma \\ 0, \text{Model X-ray}(C_\theta) > \gamma. \end{cases} \quad (5)$$

As shown in Fig. 9 and Fig. 10, for the same dataset (taking CIFAR-10 as an example), we find that the realtionship of **RE** and **ATS** between clean and backdoor models exhibits consistency. This allows us to determine an estimated threshold $\bar{\gamma}$ based on a small set of models:

$$\bar{\gamma} = \frac{1}{N} \sum_{m=1}^{N} \arg\max_{\gamma \in \Gamma} \frac{2 \times \left( \text{precision}_\gamma \times \text{recall}_\gamma \right)}{\left( \text{precision}_\gamma + \text{recall}_\gamma \right)}. \quad (6)$$

Based on thresholds $\bar{\gamma}$ (*e.g.*, for **Ours-RE** CIFAR-10: 0.873, GTSRB: 2.040, CIFAR-100: 1.194; for **Ours-ATS**, CIFAR-10: 0.184, GTSRB: 0.134, CIFAR-100: 0.040), the detection accuracy on CIFAR-10 is 87.5%, on GTSRB is 93.75% and on CIFAR-100 is 100%. *Model X-ray* consistently identifies anomalies in the decision boundaries that three samples are encircled by a large area of the target label, demonstrating precise detection of backdoored models and determine the attack target labels. The visualized decision boundaries can be found in the supplementary material.

## 5.4 The Efficiency of *Model X-ray*

Neural Cleanse and MM-BD necessitate access to the model's parameters, and MNTD relies on logit outputs from the target model. *Model X-ray* detects the backdoor model solely by predicted hard labels of clean inputs from the model. In Table 4, we show the number of benign samples that the defender needs. Both Neural Cleanse and MNTD necessitate a certain proportion of benign data (*e.g.*, 5% of the benign dataset) to complement their defense mechanisms, MM-BD does not require any clean data. Our method necessitates only three benign samples to plot a decision boundary, and with $N$ set to 20, only 60 clean samples are required, which is already sufficient to ensure the effectiveness of our detection.

In addition, we compare the average inference time of each method in Table 5. The experiment is conducted on one NVIDIA RTX A6000. Specifically, Neural Cleanse requires a trigger reverse engineering optimization process for each class, MM-BD also requires a margin statistical process to obtain a maximum margin statistic for each class, and MNTD requires preparation that generates a large set of shadow models (1024 clean models and 1024 attack models) to train a meta-classifier. In contrast, our method eliminates the need for any optimization or training processes, making it a versatile plug-and-play solution that functions as a lightweight diagnostic scanning tool.

Table 4: Benign samples required for different methods.

| Method | CIFAR-10 | GTSRB | CIFAR-100 | ImageNet-10 |
|---|---|---|---|---|
| Neural Cleanse | 2500 | 1332 | 2500 | 473 |
| MNTD | 2500 | 1332 | 2500 | 473 |
| MM-BD | 0 | 0 | 0 | 0 |
| Ours | 60 | 60 | 60 | 60 |

Table 5: The average inference time(sec) for different methods. † means the training time(sec).

| Method | Neural Cleanse | MNTD † | MNTD | MM-BD | Ours |
|---|---|---|---|---|---|
| CIFAR-10 | 243.4 | 44268.6 | 0.06 | 75.2 | 36.5 |
| GTSRB | 628.5 | 53409.0 | 0.05 | 334.8 | 34.6 |
| CIFAR-100 | 2431.7 | 46680.9 | 0.06 | 829.5 | 36.0 |
| ImageNet-10 | 1471.0 | 73632.2 | 1.5 | 414.4 | 112.3 |

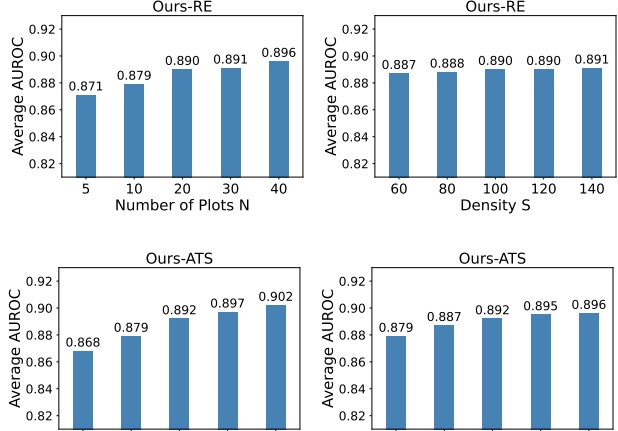

Figure 12: The influence of the number of plots $N$ and point density $S$.

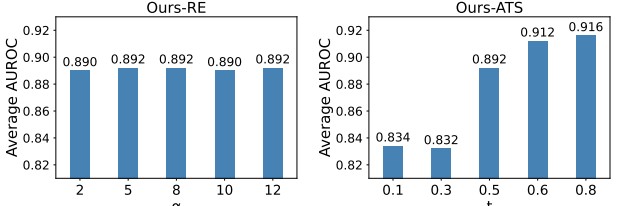

Figure 13: The influence of the parameters $\alpha$ and $t$.

## 5.5 Ablation Study

**The Influence of the Hyper-parameters.** $N$ is the number of decision boundary plots and $S$ is the density of decision boundaries, which are critical to the evaluation efficiency. Here, we investigate *Model X-ray*'s performance under fixed $N = 20$ with $S$ ranging from 60 to 140 and under fixed $S = 100$ with $N$ ranging from 5 to 40. Fig. 12 shows that lower $N$ and $S$ will slightly degrade the performance of *Model X-ray* on CIFAR-10, which is still acceptable.

Besides, we investigate the impact of parameters in two indicators, *i.e.*, $\alpha$ in RE and $t$ in ATS. As shown in Fig. 13, different

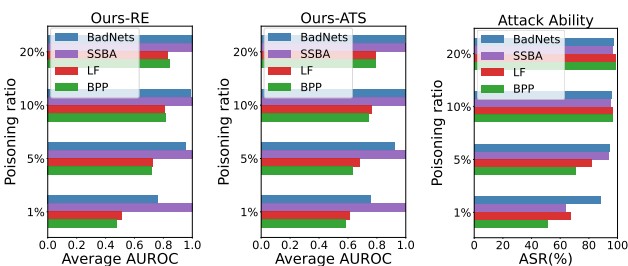

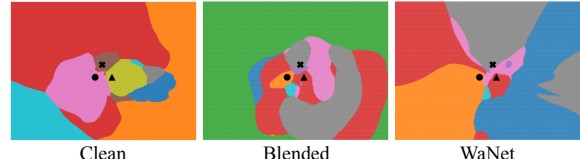

Figure 14: The influence of the poisoning ratio.



Figure 15: Decision boundaries of Blended [10] and WaNet [34].

$\alpha$ has a neglectable effect on **Ours-RE**, while $t$ larger than 0.5 is better for **Ours-ATS**.

**The Influence of the Poisoning Ratio.** In the above experiment, we set the poisoning ratio as 10% by default. Here, we further evaluate our method against data-poisoning attacks under different poisoning ratios (1%, 5%, 10%, and 20%) on CIFAR-10 dataset. As shown in Fig. 14, as the poisoning ratio increases, our approach becomes more effective, indicating that the phenomenon of anomalous decision boundaries in the backdoor models becomes more pronounced. For low ratios like 1%, the attack ability for some attacks degrades, wherein the poorer performance is understood.

## 6 DISCUSSION

**Special Cases.** We find that **Ours-AST** can distinguish the backdoored model by WaNet [34] from the clean model. Differently, the **AST** of WaNet is larger rather than smaller than that of the clean model (see Fig. 15). We conjecture that WaNet can be seen as an augmentation enhancing the robustness of clean samples. Blended [10] can bypass our detection. We explain that blending the trigger pattern with clean samples may not establish the shortcuts because of the redundancy of the model, which can be easily purified by pruning like ANP [47]. Nonetheless, we need more sophisticated strategies to achieve better detection.

## 7 CONCLUSION

In this paper, we present a noteworthy observation: there exists a distinction between clean models and backdoored models by visualized 2D decision boundaries. Based on this, we propose *Model X-ray*, a novel post-training backdoor detection approach through the analysis of illustrated 2D decision boundaries, which solely relies on the hard prediction of clean inputs, regardless of any assumptions about backdoor attacks and can determine the target label under the all-to-one attack strategy.

Extensive experiments support that *Model X-ray* has outstanding effectiveness and efficiency against diverse backdoor attacks on different datasets and different architectures.

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
