# OpenReview forum: "Model X-ray : Detecting Backdoored Models via Decision Boundary"
_acmmm.org/ACMMM/2024/Conference — MM2024 Poster_

### Official Review · Reviewer_RDGn · 2024-05-16

**Rating:** 4
**Confidence:** 3

**Summary:**

In this paper, the authors present an observation that the visualized 2D decision boundaries between clean models and backdoored models can be distinguished. Based on this, they propose Model X-ray, a post-training backdoor detection approach through the analysis of illustrated 2D decision boundaries. The performance of Model X-ray under various scenarios is impressive. The paper is well written.

**Strengths:**

• This paper presents a noteworthy observation that backdoor models have significant different decision boundary with benign models.

  • Model X-ray does not need any assumptions about backdoor attacks and can determine the target label under the all-to-one attack strategy.

  • Good visualization about decision boundary. Two decision boundary-based methods are provided for backdoor detection, which are Rényi Entropy (RE) and Areas dominated by Triple Samples (ATS). Results demonstrate Model X-ray is effective in detecting backdoor models on different datasets, model architectures, and backdoor attacks.

**Limitations:**

• Some important explanations are missing in the paper. For example, the reason why the two strategies were chosen is not illustrated in section 4.2. The experiments on different datasets on the same model architecture are not provided in the paper.

  • The explanation of some results is a bit exaggerated. For example, in Figure 7-8 the backdoored models do not have significantly smaller RE/ATS than clean models. In addition, multiple results in evaluation lack explanation or are presented informally.

  • More discussions on the weakness of Model X-ray are needed.

**Questions:**

  • In Section 5.4, the authors highlighted the efficiency of Model X-ray which requires a small number of clean samples. I am curious about the relationship between the number of clean samples and the performance of Model X-ray. Will more clean samples affect the decision boundaries?

  • The experiments on different datasets on the same model architecture were not provided in the paper. Can you explain why? In addition, in Fig. 5 (left), should p3 be equal to A(3) / A(B_k) based on Formula 2?

**Suitability:**

3

---

### Official Review · Reviewer_3o4p · 2024-05-24

**Rating:** 4
**Confidence:** 4

**Summary:**

The paper introduces Model X-ray, a post-training backdoor detection method based on decision boundary analysis.The method is based on the interesting observation that the decision boundaries of backdoor models exhibit a greater degree of proximity than the decision boundaries of clean models. Therefore, Model X-ray devises two metrics to capture this difference and detect backdoor models. Experiments show that Model X-ray outperforms existing methods in terms of detection accuracy and efficiency.

**Strengths:**

1. The paper is well organized and easy to understand.
2. The proposed method is novel and provides a new perspective for backdoor detection.

**Limitations:**

The main concern is the evaluation of the proposed method, especially the selection of baselines:

1. For backdoor detection, this paper adopts Neural Cleanse, MNTD and MM-BD as the baseline, which represent three different types of defenses. Neural Cleanse is the first backdoor detection method based on trigger inversion, while the recent trigger-inversion-based methods have achieved much better performance. Therefore, it is necessary to explain the reason for choosing NC in this paper.

2. As claimed in discussion, Blended can bypass the proposed detection method. While it should not be required that the proposed method be valid for all attacks, there should be more validation and analytical description of the invalid cases. In particular, there are a series of backdoor attack methods similar to Blended.

3. Can attackers build up adaptive attacks to bypass the proposed method? This is important for a more comprehensive assessment of the robustness of the proposed method.

**Suitability:**

2

---

### Official Review · Reviewer_G3ZM · 2024-06-05

**Rating:** 3
**Confidence:** 3

**Summary:**

The paper "Model X-ray: Detecting Backdoored Models via Decision Boundary" presents a novel approach to identifying backdoored deep neural networks (DNNs) by analyzing decision boundaries. The proposed method, termed Model X-ray, leverages 2D decision boundary visualizations to distinguish between clean and backdoored models. The technique is based on two metrics: Rényi Entropy (RE) and Areas Dominated by Triple Samples (ATS). Extensive experiments demonstrate the effectiveness and efficiency of Model X-ray across various backdoor attacks, image datasets, and architectures.

**Strengths:**

1. The use of 2D decision boundary visualizations to detect backdoored models is a novel and interpretable method that adds significant value to the existing body of work in backdoor detection.
2. Model X-ray is designed to work in a post-training scenario without requiring knowledge of the attack information, logit output, or model parameters, making it highly practical and applicable in real-world scenarios.
3. The paper provides a thorough comparison with existing post-training detection methods such as Neural Cleanse, MNTD, and MM-BD, demonstrating superior performance in most cases.

**Limitations:**

1. The paper contains several typos, which detract from its overall readability and professionalism. Specifically:
 Page 2: "phonemona" should be "phenomena".
 Page 2: "Besdies" should be "Besides".
 Page 7: "Speifically" should be "Specifically".
 Page 7: "realtionship" should be "relationship".
2. The paper compares the proposed method with Neural Cleanse, MNTD, and MM-BD. However, it lacks comparisons with some of the latest state-of-the-art methods published after 2022, such as ASSET, BIRD, and other advanced detection techniques. Including these comparisons would provide a more comprehensive evaluation of the method's effectiveness.
[1] ASSET: Robust Backdoor Data Detection Across a Multiplicity of Deep Learning Paradigms
[2] BIRD: Generalizable Backdoor Detection and Removal for Deep Reinforcement Learning
3. The robustness of the proposed method against different levels of attack severity, variations in trigger patterns, and noise is not thoroughly analyzed. Experiments focusing on these aspects would provide a deeper understanding of the method's strengths and limitations.
4. The approach lacks formal theoretical guarantees regarding its detection capabilities. The conditions under which the decision boundary visualization reliably distinguishes between clean and backdoored models are not rigorously defined, leading to potential uncertainties in its effectiveness.
5. The effectiveness of decision boundary-based detection may vary across different domains and types of data. The method's generalizability to diverse application domains, such as natural language processing, medical imaging, or time-series data, is not extensively validated.

**Suitability:**

2

---

### Meta-Review · Area_Chair_WsBD · 2024-07-01

**Recommendation:** Accept (Poster)
**Confidence:** 5

**Metareview:**

A rebuttal letter is submitted. All reviewers consider this work has contributed to the filed of identifying backdoored deep neural networks. A decision of acceptance is converged by three reviewers, and the authors are requested to incorporate the rebuttal letter in final submission. Thus, I recommend acceptance.